# Impact of COVID-19 Pandemic on Quality of Health Services Provided by Emergency Medical Services and Emergency Departments in the Opinion of Patients: Pilot Study

**DOI:** 10.3390/ijerph19031232

**Published:** 2022-01-22

**Authors:** Piotr Konrad Leszczyński, Paulina Sobolewska, Bożena Muraczyńska, Paulina Gryz, Adam Kwapisz

**Affiliations:** 1Faculty of Medical Sciences and Health Sciences, University of Natural Sciences and Humanities, 08-110 Siedlce, Poland; bozena.muraczynska@uph.edu.pl (B.M.); paulina122v@gmail.com (P.G.); 2Clinic of Orthopedics and Pediatric Orthopedics, Medical University of Łódź, 90-419 Łódź, Poland; brodzinskapaulina@gmail.com (P.S.); adam.kwapisz@gmail.com (A.K.)

**Keywords:** COVID-19, emergency medical service, emergency department, health services, patients’ opinion

## Abstract

Background: The aim of this study was to assess the quality of assistance provided by emergency medical service (EMS) and hospital emergency department (ED) staff in the opinion of patients. Methods: Simple random sampling was performed to select the responders. A group of 131 people who correctly completed the questionnaire form was included in the study. The questionnaire was directed only to patients receiving EMS and/or ED assistance during the time period when the pandemic was announced in Poland. Results: Problems in the ED were reported by 38 people (31.15%), and calling EMS was reported as generating problems by 7 people (16.67%). Dissatisfaction with the help provided in EDs was reported by 33 patients (27.05%), and dissatisfaction with the help by EMS was reported by eight people (19.05%). The assessment of the patient’s satisfaction level significantly correlated with the waiting time in the ED (Spearman = 0.217; *p* = 0.016) as well as with the waiting for EMSs (Spearman = 0.403; *p* = 0.008). Conclusions: Patients of the ambulance service and hospital emergency departments during the pandemic positively assessed the actions of medical services, despite significant delays in providing health services.

## 1. Introduction

The Act on Patients’ Rights and Patient Ombudsman guarantees a patient’s rights to health services [1]. The legislation specifies that the patient has the right, in a situation of limited possibilities, to be provided with appropriate health services, including transparent and objective procedures based on medical criteria, determining the order of access to these services. According to Polish law [1], every person in the country should receive immediate medical assistance in life-threatening situations. The COVID-19 pandemic has exposed problems related to health care in Poland, and demonstrated the weakness of the system in terms of patients’ access to health services. The growing number of cases of severe acute respiratory syndrome (SARS-CoV-2) infections overburdened the healthcare systems, which have limited human resources, inevitably threatening the continuity of provision of healthcare services in health and life-threatening situations. The duration of the COVID-19 pandemic has caused queues and the time associated with waiting for health services by patients to be extended [2]. The functioning of medical entities has changed significantly. At the beginning of March 2020, specialist hospitals for COVID-19 patients were established, planned admissions were limited, as were the medical appointments to receive primary healthcare services and outpatient specialist care [3]. Services provided by general practitioners were mostly limited to tele-advice (telephone consultations) [4]. Temporary hospitals and special COVID-19 wards have also been established, and many healthcare professionals have been directed to care for patients suffering from SARS-CoV-2. Along with the implementation of the National Immunization Program, there was a need to arrange additional staff to qualify patients and perform vaccinations [5]. Due to changes related to the pandemic, increased responsibilities, and pre-existing staff shortages, healthcare workers found themselves in a difficult situation, which had an impact on their mental health [6]. 

The impact of the COVID-19 pandemic on the quality of health services is clearly noticeable. In the scientific literature, numerous examples point to problems with the availability of medical services, as well as patients’ fear of becoming infected during a visit to a medical facility. Analyzing the time before its occurrence, the frequency of receiving emergency aid has significantly changed [7]. At first, emergency wards were overcrowded due to limited access to GPs [8]. With the development of the pandemic and the posting of messages communicating the need to stay at home and maintain social distancing, the number of medical visits and appointments decreased sharply. This can be explained by social fear about the possibility of infection and the sense of responsibility for other people who needed help more [9,10]. The downward trend also correlated with injury-inflicted medical visits, as patients showed fear of not receiving medical help due to the reduced numbers of active healthcare professionals. The number of road accidents also decreased during that time [11]. The data in connection with seeking help by people in a life-threatening condition, but not due to the SARS-CoV-2 infection, appear ominous. This is true, for example, in the case of people who experience stroke symptoms, where every minute is crucial for the medical outcome. During the COVID-19 pandemic, these patients, despite the occurrence of clear symptoms, did not reach out for immediate medical help, and even avoided seeking help altogether [12]. Despite a pronounced decrease in seeking emergency care, cases related to mental disorders and alcohol abuse are increasing [13,14]. The reasons mentioned above are only examples of a long list of factors influencing the quality of healthcare services during a pandemic. In our opinion, the above-mentioned behavior of patients and the disorganization of medical services may have significantly affected the quality of services in emergency medical services.

Our aim in this study was to develop a general assessment of patient satisfaction and of the availability of emergency care in hospital emergency departments (EDs) and of emergency medical services (EMSs) since the start of the COVID-19 pandemic in Poland. 

## 2. Materials and Methods

This research was conducted on a sample of 150 people who, during the COVID-19 pandemic, received emergency assistance from hospital emergency departments (EDs) and emergency medical services (EMSs). The study inclusion criterion was the 100% completion of a questionnaire for at least one of the aspects (ED or EMS patient). In total, 131 respondents’ responses were eligible for analysis. This research was conducted as cooperation between two universities (Łódź and Siedlce) in central Poland from April to September 2021, using a questionnaire purposefully designed and prepared in both electronic and printed versions. The survey contained a total of 25 questions, including questions about socio-demographic data. The questions asked of respondents included in the survey were single-choice, semi-open, and open-ended questions. 

The study group was selected only from among ED and EMS patients who personally received emergency care since the start of the pandemic in Poland. Due to the ongoing pandemic during the study, we decided to send the questionnaires remotely, including via the Internet. The respondents did not receive any profit for responding. The survey was posted on websites for self-completion by respondents and displayed automatically when they accessed any website (e.g., pop-up format). The questionnaire was anonymous and did not allow for the identification of people participating in the study. The participants provided their informed consent to participate in the survey, and the methodology was reported to the relevant bioethics committee (No. 11/2011, University of Siedlce). Due to the method of obtaining material for research and the anonymity of the study, voluntary completion of the questionnaire was tantamount to consenting to participation in the study. 

The acquired data were statistically analyzed using Statistica PSPP and Statistica 13. The chi-squared test and the Mann–Whitney U test were used to check the frequency of reporting problems during admission according to the patient’s sex, age, and method of admission. The Mann–Whitney U test was also used to test the relationship between admission method and waiting time. The correlation between the level of patient satisfaction and the waiting time or patient’s place of residence is described by the Pearson correlation coefficient.

## 3. Results

### 3.1. Characteristics of The Study Group

A total of 131 respondents correctly completed the questionnaire, including 86 women (65.65%) and 45 men (34.35%). The mean age was 36.32 years (SD ± 12.54 years). Most respondents had higher education (*n* = 86; 65.65%) or secondary education (*n* = 40; 30.53%). The place of residence of the respondents was indicated as follows: rural (*n* = 71; 54.20%), town up to 50,000 inhabitants (*n* = 33; 25.19%), and town up to 100,000 inhabitants (*n* = 17; 12.98%). Patients mostly originated from the following voivodeships: Lubelskie (*n* = 64; 48.85%), Mazowieckie (*n* = 48; 36.64%), Wielkopolskie (*n* = 8; 6.11%), and Łódzkie (*n* = 7; 5.34%). 

Since the start of the COVID-19 pandemic in Poland, 122 (93.13%) respondents received hospital emergency department assistance, and 42 (32.06%) received EMSs. In 33 cases, patients received both EMS and ED assistance. 

### 3.2. Interventions in The EDs

The respondents received help in the ED on average 1.32 times (SD ± 0.67). The most common reasons were sudden illness (*n* = 63; 51.64%) and accidents/injuries (*n* = 47; 38.52%). Other reasons for reporting to the ED were symptoms related to COVID-19 (*n* = 6; 4.92%), no help received from a primary care physician (*n* = 4; 3.28%), and exacerbation of a chronic disease (*n* = 2; 1.64%). Difficulties with admission to an ED were reported by 38 people (31.15% attending the ED). The main problems indicated were: “long waiting time”, “lack of places”, and “lack of sufficient interest on the part of medical staff”. The waiting time for admission to the ED and starting diagnostics in 58 cases (47.54%) exceeded 1 hour. The remaining people were admitted faster; a detailed list is presented in Table 1.

Most of the patients attended the ED with their own transport without a referral (*n* = 92; 75.41%). The remaining patients were referred by their general practitioner (GP; *n* = 19; 15.57%) or were brought in by an emergency medical service (*n* = 11; 9.02%). Almost half of the respondents (*n* = 58; 47.54%) waited for over an hour to be admitted. Only 16 patients (13.11%) were consulted within 15 min of waiting, 18 (14.75%) within 30 min of waiting, 10 (8.20%) within 45 min, and 20 patients (16.39%) within 1 hour of waiting. Due to the long waiting time, 30 people (24.59%) resigned from receiving medical assistance in the ED. As a result, the most of respondents were discharged home (*n* = 100; 81.97%), 16 people (13.11%) were admitted to hospitalization, and six patients (4.92%) were transferred to another medical entity. 

Most of the patients were satisfied with the help provided in the ED (*n* = 68; 55.74%). Thirty-three (27.05%) were dissatisfied. The remainder expressed no opinion. A detailed breakdown of the responses expressing ED patient satisfaction is presented in Figure 1 and Figure 2 (Likert scale: very unsatisfied, −2; unsatisfied, −1; neutral, 0; satisfied, +1; very satisfied, +2).

### 3.3. Correlations of Data of ED Patients

We assessed the factors that may have impacted the access to emergency aid during the pandemic in Poland. Problems during admission to EDs occurred almost as often among men (26.19%) as among women (33.75%) (χ^2^(1, *N* = 122) = 0.734, *p* = 0.392), and the frequency of reporting problems was not related to the age of the patient (U (*N* _reporting problem_ = 38, *N* _not reporting problem_ = 84) = 1361.50, Z = 1.29, *p* = 0.196). The problems reported by the patients were also not correlated with a type of means of transport to the ED: troubles were reported by 30.45% patients who attended to the ward on their own, by 26.32% patients who attended based on a referral from a doctor, and by 45.45% patients brought by EMSs (χ^2^(1, *N* = 122) = 1.279, *p* = 0.528). The waiting time in the ED was independent of whether the patient arrived independently or was brought by the EMS (U (*N* _by their own_ = 92, *N* _brought by EMS_ = 11) = 392.00, Z = 1.21, *p* = 0.226).

The assessment of the patient’s satisfaction level significantly correlated with the waiting time in the ED (r = −0.217; *p* = 0.016; Figure 3). The longer the waiting time, the lower the patient’s satisfaction. A significant relationship was also observed in the correlation of the patient’s place of residence with the level of patient’s satisfaction with the help provided in the ED (r = −0.276; *p* = 0.002). People from larger towns showed a tendency to be more critical of ED staff than people from suburban agglomerations. 

### 3.4. EMS Interventions

The respondents received EMS assistance on average 1.39 times (SD ± 0.82). The most common reasons were sudden illness (*n* = 14; 33.33%) and accidents/injuries (*n* = 14; 33.33%). Other reasons for calling EMSs were symptoms related to COVID-19 (*n* = 8; 19.05%), exacerbation of a chronic disease (*n* = 4; 9.52%), as well as no help received from a primary care physician (*n* = 2; 4.76%). Difficulties with calling EMS were reported by seven people (16.67%). The main problems indicated were: “long waiting time”, “lack of a free ambulance”, and “lack of courtesy on the part of the medical dispatcher”. The waiting time for an ambulance exceeded 1 hour in seven cases (16.67%). The remaining people waited for EMSs for a shorter period (a detailed list is presented in Table 1). Seven people (16.67%) were refused by the medical dispatcher when calling for an ambulance. Due to the long waiting time, 10 (23.81%) people gave up on calling the EMS. The same number of respondents gave up on calling EMSs at least once during the pandemic for fear of infection with SARS-CoV-2. The majority of the respondents (*n* = 24; 57.14%) were not taken to hospital, and only in three cases were drugs administered on the spot. Eighteen people (42.86%) needed to be transported to hospital.

Most of the patients were satisfied with the help provided by the EMS (*n* = 29; 69.05%); eight people (19.05%) were dissatisfied. The remainder expressed no opinion. A detailed breakdown of the responses regarding the satisfaction of EMS patients is presented in Figure 4 (Likert scale: very unsatisfied, −2; unsatisfied, −1; neutral, 0; satisfied, +1; very satisfied, +2).

### 3.5. Correlations of Data of EMS Patients

We found a statistically significant correlation between the patients’ sex and the occurrence of problems when calling EMS: 4.17% of women and 33.33% of men reported problems (χ^2^(1, *N* = 42) = 6.300, *p* = 0.012). However, this situation did not correlate with the age of the calling persons (U(*N* _reporting problem_ = 7, *N* _not reporting problem_ = 35) = 80.00, Z = 1.43, *p* = 0.151). The waiting time for EMSs as be independent of the type of event (χ^2^(12, *N* = 42) = 10.81, *p* = 0.545) and of the patient’s place of residence (χ^2^(12, *N* = 42) = 17.20, *p* = 0.142).

The patient’s satisfaction level significantly correlated with the waiting time for the EMSs (r = -0.403; *p* = 0.008; Figure 5). The longer the waiting time, the lower the patient’s satisfaction. As in the case of ED patients, no correlation of the level of satisfaction with the place of residence of the respondents was found (r = 0.061; *p* = 0.703). 

## 4. Discussion

The most important finding of our study is that COVID-19 pandemic negatively affected the functioning of the healthcare system in all areas in Poland [15,16,17]. Patients’ access to health services was limited, and long waiting times resulted in patients not seeking help. This difficult period revealed the shortcomings that exist in many public health sectors and a complete lack of preparedness for emerging challenges and problems in helping patients. [18]. However, the number of EMS calls and admissions to EDs decreased when comparing this time to the time prior to the pandemic [19].

The majority of respondents (*n* = 122; 93.13%) received ED assistance, most often reporting to a hospital due to a sudden illness (*n* = 63; 51.64%) or an accident or injury (*n* = 47; 38.52%). These are the first places attended by patients in a state of health hazard. Notably, apart from the daily patients reporting with COVID-19, there are also patients reporting with symptoms of SARS-CoV-2. This has extended the waiting time for help and the further provision of healthcare services for patients [20]. In 58 cases (47.54%), the admission time to the ED exceeded 1 hour, related to the difficulties resulting from the overburdening of the healthcare system. Only a thorough analysis of the functioning of this type of ward and the quick reaction of decision-makers in response to changing conditions can help face this type of problem given the availability of healthcare assistance [21,22]. Despite the waiting time being longer than before the pandemic, most respondents expressed their satisfaction with the help they received at the EDs (*n* = 68; 55.74%). 

Among the respondents, 42 people received EMS help. The respondents indicated sudden illness or accident or injury as the reason for calling EMSs (*n* = 14; 33.33%). Due to the uniqueness of the pandemic situation that burdened all healthcare systems, the waiting time was longer, and in seven cases (16.67%), it exceeded an hour, which was also indicated by patients as a problem when calling the EMS, apart from the lack of an available ambulance and “no courtesy showed on the part of the medical dispatcher”. Due to the long waiting time, 10 people (23.81%) gave up on calling EMSs; these people may have used their own transport to receive medical help or gave up on it completely. EMS staff are obliged to disinfect themselves personally and their ambulance after each call due to the possibility of virus transmission. This is a time-consuming procedure, which causes the EMS to be withdrawn from operation for about 1 hour [23]. Despite these difficulties, most patients expressed their satisfaction with the help provided by EMSs (*n* = 29; 69.05%). 

Both in the case of assistance provided at the ED or by EMSs, the level of patient satisfaction significantly correlated with the waiting time for help: the shorter the waiting time, the higher the patient’s satisfaction with the services provided. Moreover, patients from cities showed a lower level of satisfaction with ED medical services compared to the respondents from suburban agglomerations. The level of expectations of city dwellers regarding the level of health services is probably higher; therefore, we found a significant correlation in this aspect. This issue should be assessed in more detail in future targeted research. According to our current knowledge, this study is the first analysis of the quality of health services provided by the emergency healthcare system based on the opinion of patients during the COVID-19 pandemic. At the time of writing this study, there were only isolated reports in studies of patient satiety. Sharma et al. (2021) conducted an analysis of patients with a positive COVID-19 test, indicating their high satisfaction during hospitalization in a third reference hospital [24]. Jannati et al. (2021) proved that patients who used teleconsultation during the pandemic were mostly satisfied with the provided services [25].

Life-threatening health services can cause a patient to experience fear, frustration, and anger. Patients are then demanding of medical personnel [26]. We were aware of these problems; therefore, we undertook a difficult pilot study on ED and EMS patients. Research should continue to detail the elements of the healthcare system that can be more prepared for future pandemic events of a similar nature. 

Considering the results of this study and the analysis of the available literature, we suggest several solutions for the future:(1)Information sessions should be conducted. in society on the functioning of EDs and EMSs. The problem of unjustified ambulance calls and ED visits has long been known, so programs should be implemented to inform patients where to find help in non-life-threatening cases.(2)Transparent systems of patient segregation upon admission to ED should be implemented, indicating the real waiting time for the start of diagnostics (e.g., supported by a computer program and artificial intelligence). Patients should understand the therapeutic priority given by ED personnel.(3)Ambulance services should be reorganized with the division of teams depending on the purpose (e.g., resuscitation, reconnaissance, transport, etc.). Urgent cases can then be dealt with immediately and patients who can expect to wait longer will receive initial diagnosis and treatment at home. Separate ambulances may be used to transport patients to the hospital for diagnostics if their condition is not life-threatening.

### Study Limitations

Due to the ongoing pandemic and the sanitary regime imposed, we decided to conduct this study remotely, which limited the possibility of extending the questionnaire to an additional oral interview. However, the provision of somewhat more information was made possible by answering open-ended questions. Patients of the Polish Airborne Ambulance Service were not included in the study due to the distinct nature of the work of HEMS, both during the pandemic and before its start [27,28,29,30]. Moreover, it is difficult to determine the size of the representative group due to the lack of aggregate data on the national scale of the population of people attending emergency wards and receiving EMSs. The small sample size may be due to the reluctance of EMS and ED patients to reassess their traumatic events. Moreover, in Poland, there is no system for assessing the quality of health services; therefore, patients do not believe that medical services will be improved by means of questionnaires. The negative patient feedback identified in the study may be due to reasons other than the COVID-19 pandemic.

## 5. Conclusions

During the pandemic, patients reported significant delays in accessing help from EMSs and EDs. The opinion on the quality of health services was positive in most cases, and the assessment significantly correlated with cases of long waiting times. The characteristics of the incidents resulting in calls for ambulance service and reporting to emergency wards during the pandemic were determined. 

## Figures and Tables

**Figure 1 ijerph-19-01232-f001:**
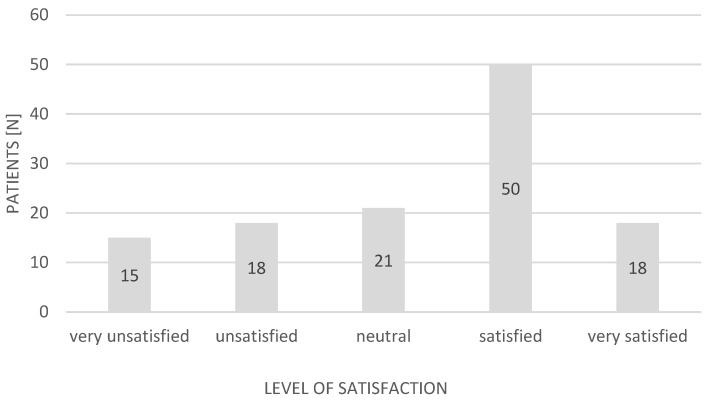
Patients’ satisfaction with the help provided in the ED during the pandemic (Likert scale).

**Figure 2 ijerph-19-01232-f002:**
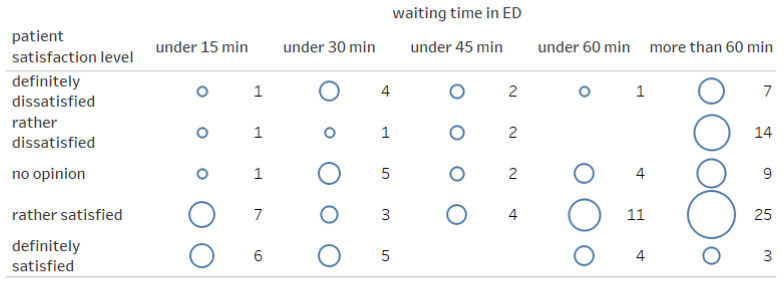
Patient satisfaction depending on the waiting time in the ED.

**Figure 3 ijerph-19-01232-f003:**
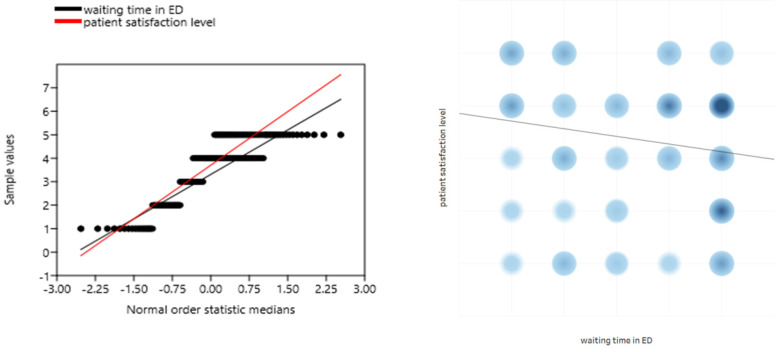
Correlation of the level of patient satisfaction with the waiting time in the ED.

**Figure 4 ijerph-19-01232-f004:**
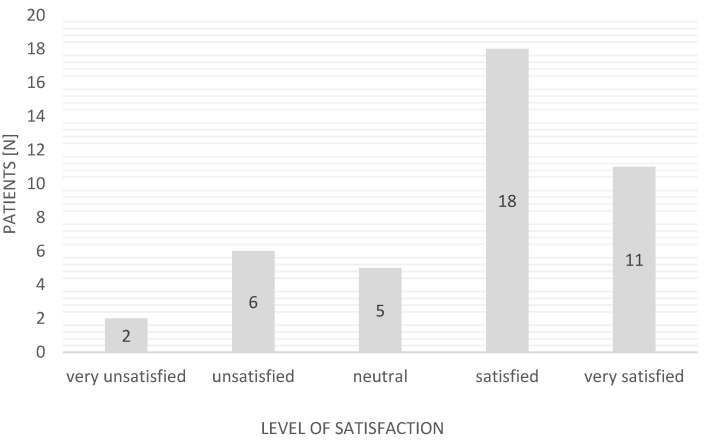
Patients’ satisfaction with the help provided by EMSs during the pandemic (Likert scale).

**Figure 5 ijerph-19-01232-f005:**
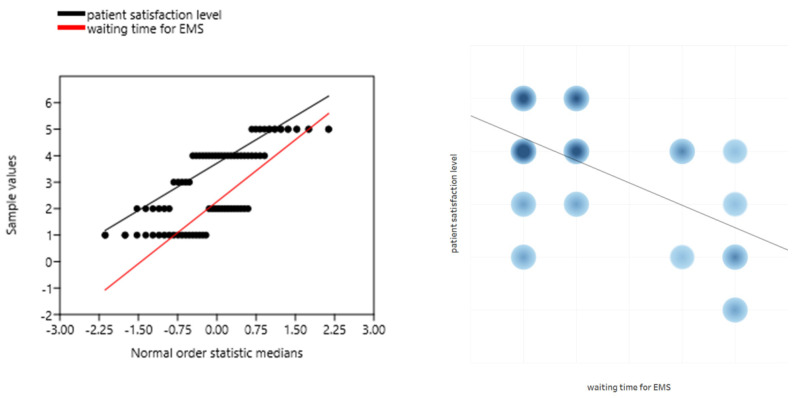
Correlation of the level of patient satisfaction with the waiting time for EMSs.

**Table 1 ijerph-19-01232-t001:** Waiting time for admission to the ED.

Waiting Time	Patients, *n* (%)	
	ED	EMS
Up to 15 min	16 (13.11%)	18 (42.86%)
Up to 30 min	18 (14.76%)	13 (30.95%)
Up to 45 min	10 (8.20%)	0 (0.00%)
Up to 60 min	20 (16.39%)	4 (9.52%)
Over 60 min	58 (47.54%)	7 (16.67%)
Total	122 (100.00%)	42 (100.00%)

## Data Availability

Not applicable.

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
