# Peer review of "Impact of COVID-19 Pandemic on Quality of Health Services Provided by Emergency Medical Services and Emergency Departments in the Opinion of Patients: Pilot Study"

_ijerph, 2022, doi:10.3390/ijerph19031232_

Round 1

Reviewer 1 Report

Dear authors

At first, I appreciated the modifications to the original manuscript, especially for statistical analysis. Here, I reported my concerns about the manuscript in the present form.

Introduction: This section requires further modification, especially from lines 67 to 85. I found this paragraph not related to your topic. I suggest you review this part and adapt it to the study's primary purpose.

Material and section methods: At line 93, you stated that 131 respondents met the inclusion criteria, without mentioning what criteria you used to include/exclude subjects. I suggest reporting this detail in the result section.

From lines 99 to 101, you repeat how the questionnaire is composed (see lines 97-99).

You stated that you performed logistic regression analysis....well! Where are the results???? I never read "odds ratio" or similar concepts in the results.

Results: On page 5, paragraph 3.3 was confusing and difficult to read. I sincerely did not understand the results provided by your statistical analysis. Furthermore, you used a new abbreviation never specified (i.e., "U"?).

Discussion: In the result section, line 181, you reported that "people from largertowns showed a tendency to be more critical of ED staff than people from suburban agglomerations". How did you interpret this result? 

In conclusion, despite the interesting topic, I found critical issues. Therefore, I invite the authors to provide detailed results according to the statistical methods used.

Author Response

Thank you for your valuable comments. All authors made appropriate corrections as indicated by the reviewer:

1) Introduction - has been completed and clarified, especially the paragraph from lines 67 to 85.

2) Material and section methods - the inclusion criteria have been supplemented and explained. The description of the statistical methods used in the study was also improved so that it was adequate to the results.

3) Repeated phrase From lines 99 to 101 has been deleted.

4) Results - subchapter 3.3 was corrected. The description of the statistical analysis was reported according to the APA standard (e.g. standard U-factor).

5) Discussion - explained the etiology of the result describing that "people from larger cities tended to be more critical of the staff of the emergency department than people from suburban agglomerations".

6) The entire text has been re-edited linguistically by ordering a professional paid service from the MDPI publishing house.

Reviewer 2 Report

I really appreciate authors' efforts in improving their paper. However, many of the previous flaws remains, including methods section too poor, need of extensive editing of English language and style throughout the whole paper. For the Discussion, emerging evidence has been highlighted the impact of COVID-19 on healthcare services and facilities (including ER): authors should use this body of evidence to compare results and suggest future perspective. In brief, this paper is not suitable for publication as presented. 

Author Response

Thank you for your valuable comments. All authors made appropriate corrections as indicated by the reviewer:

1) Introduction - has been completed and clarified, especially the paragraph from lines 67 to 85.

2) Material and section methods - the inclusion criteria have been supplemented and explained. The description of the statistical methods used in the study was also improved so that it was adequate to the results.

3) Repeated phrase From lines 99 to 101 has been deleted.

4) Results - subchapter 3.3 was corrected. The description of the statistical analysis was reported according to the APA standard (e.g. standard U-factor).

5) Discussion - explained the etiology of the result describing that "people from larger cities tended to be more critical of the staff of the emergency department than people from suburban agglomerations". Proposed solutions to the problem in the future are also described.

6) The entire text has been re-edited linguistically by ordering a professional paid service from the MDPI publishing house.

This manuscript is a resubmission of an earlier submission. The following is a list of the peer review reports and author responses from that submission.

Round 1

Reviewer 1 Report

I received the ms titled “The impact of the COVID-19 pandemic on the quality of health services provided by emergency medical services teams (EMS) and hospital emergency departments (HED) in the opinion of patients”. Overall, presents a number of flaws, which have to be addressed before considering it for publication. At this stage, I cannot recommend the paper for publication in its present form. As a suggestion, authors may want to have their paper re-written for clarity.

Please, find here main comments/suggestions.

The first important concern is attributable to participants sampling and low sample size, which is limited as a real-world study and prevents from generalize findings. The paper should be considered in specific venue as a pilot study of validated the used research instrument.

Abstract

The abstract seems to exceed the IJERPH word limit. Again, it must be presented as a single paragraph without headings.

Introduction

I understand that the Act on Patients' Rights and Patient Ombudsman is a national law, but something more should be said to clarify it to we non-Polish readers.

Acronyms should be amended throughout the whole paper and at their first appearance in the text (e.g., COVID-19 in line 42; SARS-CoV-2 line 44, GP l.62, etc. and so on). Similarly, in «SARS-CoV-2 virus», “virus” is redundant since the acronym contains the word “coronavirus” (l. 71).

Authors listed some reasons for changes ER access (unavailability of GPs, decrease in road accidents, etc.) but these are not the unique reasons, so those should be presented as example.

  1. 71: hardly understand.

Methods

Methods section is limited and it is hard to understand what authors did. Data handling, analysis, statistics, etc. should be reported. Some of these information are confusingly presented in Results section, but not clear.

Complete and clear sampling strategy and respondents’ eligibility should be reported. Study period too, and in particular if it coincided with lockdown period in the country or the type of COVID-19-related restrictions that were present at the time when the study was conducted.

  1. 86 - «using a designed survey» is not clear. Was it a validated tool or a questionnaire purposely designed? I think the second, but sounds confusing. The research instrument should be included in the appendix though.
  2. 87 – socio-demographic information, not data.

Results

Without the lacking methodological information, it is hard to understand results. “Statistical analysis” sub-sections are confusing: all the analysis – although descriptive – are “statistical”. Methodological tips of analytic strategies should be presented in Methods. Results section is made for findings.

Discussion

Really poor. They should be reasons and possible explanations that would justify findings. Comparisons with other similar studies are also welcomed.

In study limitation, authors should list all the possible biases that are attributable to the type of study, survey and sampling. Similarly, the low sample size should be acknowledged as a limitation and commented. Study setting also may limit the reliability of the paper. and so on.

Author Response

Thank you for your valuable comments. All authors made appropriate corrections as indicated by the reviewer:

1) The article has been formatted as PILOT STUDY;

2) Abstract corrected according to editorial requirements;

3) The legal status in Poland was described in connection with the Act on Patients' Rights and Patient Ombudsman;

4) Throughout the work, the acronyms have been corrected and the word "virus" removed from the term SARS-CoV-2;

5) Changes in access to ED and EMS during the pandemic are given as examples, citing the relevant literature;

6) The METHODOLOGY section has been supplemented and corrected by statistics. A research tool is also described. Attached to the work of the author, who specializes in the development of methodology and statistical analysis.;

7) The RESULTS section has been completed and corrected, taking into account the correct titles of each part and the results of the statistical analysis;

8) The DISCUSSION section has been supplemented and corrected, examples of similar studies have been given, while at the same time emphasizing the innovativeness of the research carried out by the authors;

9) The limitations of the study were completed and explained in detail.

Reviewer 2 Report

Dear author
I read with interest your paper entitled "The impact of the COVID-19 pandemic on the quality of health services provided by emergency medical services teams (EMS) and hospital emergency departments (HED) in the opinion of patients."
The topic is fascinating. However, in the present form, the paper showed critical issues in detail for statistical methods used. Moreover, the article should be reviewed about language, avoiding repeating sentences (see methods section).
Here I reported my considerations. I hope that these do not disappoint you.

Introduction
The introduction was clear and well-written. The aim of the study was clearly stated.

Materials and Methods
About this section, I have some doubts, especially about two significant issues.
First, there was no mention of written informed consent or other forms, such as the local ethical committee approval or national laws to guarantee the privacy of patients. I invited the authors to provide details about this aspect.
Second, statistical methods should be reviewed. The authors should consider performing a logistic regression analysis to analyze factors related to satisfaction or dissatisfaction with emergency assistance. "Neutral" could be regarded as unsatisfied patients. Logistic regression analysis could be provided practical details, trying to analyze factors related to EMS or HED performances. I sincerely invited the authors to follow my suggestions.

As a consequence, and according to my concerns, my final decision was "major revision". 

Author Response

Thank you for your valuable comments. All authors made appropriate corrections as indicated by the reviewer:

1) The article was linguistically formatted by the ordered service of a professional Native Speaker;

2) Supplemented with explanations regarding the research tool and informed consent of the participants;

3) The METHODOLOGY and RESULTS sections have been corrected and supplemented after consulting the statistician. The statistical analysis and the correlation of the variables were explained.